# Cohort Profile: The Korean Vietnam War Veterans’ Health Study Cohort (KOVECO)

**DOI:** 10.3390/ijerph19074211

**Published:** 2022-04-01

**Authors:** Wanhyung Lee, Ui-Jin Kim, Seunghon Ham, Won-Jun Choi, Seunghyun Lee, Jin-Ha Yoon, Seong-Kyu Kang

**Affiliations:** 1Department of Occupational and Environmental Medicine, Gachon University College of Medicine, Incheon 21565, Korea; wanhyung@gmail.com (W.L.); shham@gachon.ac.kr (S.H.); wjchoi@gachon.ac.kr (W.-J.C.); 2Department of Occupational and Environmental Medicine, Gachon University Gil Medical Center, Incheon 21565, Korea; ujkim@gilhospital.com; 3Department of Preventive Medicine, Yonsei University College of Medicine, Seoul 03722, Korea; jihyun6547@naver.com (S.L.); flyinyou@gmail.com (J.-H.Y.); 4The Institute for Occupational Health, Yonsei University College of Medicine, Seoul 03722, Korea

**Keywords:** KOVECO, Agent Orange, Vietnam War

## Abstract

During the Vietnam War, many troops and citizen were exposed to large amounts of Agent Orange (AO), and the hazardous effects of AO are continuously being researched and reported. The Korean Vietnam War Veterans’ Health Study Cohort (KOVECO) is a retrospective cohort to demonstrate the health status of the Korean Vietnam War veterans and their second-generation offsprings. The KOVECO is a collaboration of data from the Ministry of Patriots and Veterans Affairs and the National Health Insurance Sharing Service from 2002 to 2018. The study participants were all Korean Vietnam War veterans and their second-generation offsprings, and the references were the general population in which gender and region were matched with the participants. As of 2002, 191,272 Vietnam War veterans (1,000,320 comparisons) and 1,963,402 s-generations (1,173,061 references) were included in the cohort. The KOVECO consists of personal information, medical facility visit information, and general health examination information. The KOVECO could act as a health surveillance system, which would be able to detect long-term health effects caused by exposure to AO and provide a direction for policy making through academic research.

## 1. Introduction

During the Vietnam War, many troops and citizens were killed or wounded and exposed to unknown threats, such as Agent Orange (AO). Large amounts of AO were widely used during the Vietnam War as a tactical herbicide without consideration of its human health effect [1]. Its hazardous effects are continuously being researched and reported [2,3,4]. Approximately more than 300,000 troops from the Republic of Korea participated in the Vietnam War. The Ministry of Patriots and Veterans Affairs (MPVA) of Korea has been rewarding veterans who participated in the war by providing medical services and welfare, which include tracking sequelae caused by AO since 1995. Epidemiologic research was performed five times in the period of 1995–2016. Although each study obtained remarkable results, they had limitations since the data used in the research were from individual research institutes. Through this study, we tried to establish a cohort with representativeness, reproducibility, and public confidence using national data.

The Korean Vietnam War Veterans’ Health Study Cohort (KOVECO) was established to perform the sixth epidemiologic study to find out the long-term health effects, in terms of unveiled sequelae, of AO on Korean Vietnam War veterans, including women troops, and the health effects on the second generation [5].

## 2. Materials and Methods

### 2.1. Cohort Description

#### 2.1.1. Data and Participants

The KOVECO is a collaboration of data from the MPVA and data from the National Health Insurance Sharing Service (NHISS). The MPVA had a list and identification information, such as name, age, sex, national identification number, family, address, and war participation information (rank, affiliation, class, date of entry into the war, or date of retirement). The MPVA defined participants of the KOVECO as extraction based on national identification number with war participation information from records accessed on 1 January 2021. The NHISS had a public database that was followed up annually on health care utilization, health screening, medical facility visit history, prescription or procedure information, sociodemographic variables, and mortality for the whole population of Korea from 2002 to 2018 [6]. The NHISS data were merged into data from the MPVA based on national identification numbers. The data merging was performed by a data expert who was independent from all researchers or relatives, and the national identification number was converted to a unique NHISS number in order to prevent individual backtracking or leakage of personal information. Data from the NHISS were collected using written informed consent from all participants, and the information was anonymized. The study was approved by the Institutional Review Board of Gachon University Gil Medical Center (IRB No. GCIRB2019-076).

The study participants included all Korean Vietnam War veterans and their second generation with an age-, region-, and sex-matched cohort. All available general populations were used as a reference group and matched by age, sex, and region with a 1:5 ratio to minimize the effects of their differences. Table 1 shows the baseline characteristics of the KOVECO in 2002. We collected information on gender and type of participants (Vietnam War veteran, second generation, and each reference). In 2002, there were 191,272 Vietnam War veterans (1,000,320 references) and 1,963,402 s-generation (1,173,061 references). The proportion of females was 0.2% of the veterans (0.2% of the reference group) and 41.1% of the second generation (40.3% of the reference group).

Table 2 shows the distribution by year of the KOVECO during the follow-up period (2002–2018). As the years passed, the overall number of participants in all groups decreased. During the follow-up period, the total person-years of Vietnam War veterans were 3,213,250 (15,329,931 person-years in the reference) and those of the second generation were 6,042,386 (38,328,109 person-years in the reference).

Table 3 shows the distribution by region of the KOVECO in 2002. In both the veteran group and the second-generation group including each reference, the number of participants is highest in Seoul (26.8% of the veteran group, 25.7% of the second-generation group), followed by Gyeonggi (15.2% and 17.8%) and Busan (9.4% and 7.7%).

#### 2.1.2. Measurement and Variables

The variables in the KOVECO are summarized in Table 4. The KOVECO consists of three databases based on a participant’s personal information, medical facility visit information, and general health examination information [7]. The personal information database contains 10 variables, including the standard year for the eligibility of the insured, the participant’s personal identification (ID), sex, birth date, district of residence, type of eligibility, percentile group of income level, grade of disability registered, type of disability registered, and type of group (Vietnam War veterans with references and second generation with references). The medical facility visit information database contains 17 variables, including the participant’s personal ID, key sequence number of claim, medical care institution identification number, first day of receiving treatment, form of claimed bill, medical subject code, main and subdisease code, first date of inpatient, route through hospitalization, days of receiving medical care and visit hospitals or inpatient, total days of prescription, extra rate applied based on claims reviewed, amount of medical expenses, expenses paid by beneficiary, and expenses paid by insurer based on claims reviewed. The general health examination information database contains 18 variables, including date of examination, the participant’s personal ID, past history (stroke, cardiovascular diseases, hypertension, diabetes, dyslipidemia, tuberculosis, and others including cancer), medical treatment history (stroke, cardiovascular diseases, hypertension, diabetes, dyslipidemia, tuberculosis, and others, including cancer), family history (stroke, cardiovascular diseases, hypertension, diabetes, dyslipidemia, tuberculosis, and others, including cancer), health behavior (smoking, alcohol consumption, and exercise level), anthropometry (height, weight, waist circumference, body mass index, visual acuity, hearing acuity), blood pressure (systolic blood pressure, diastolic blood pressure, and pulse rate), urine spot test, hemoglobin, fasting blood glucose, lipid profile (total cholesterol, triglyceride, HDL cholesterol, and calculated LDL cholesterol), serum creatinine, aspartate aminotransferase (AST), alanine aminotransferase (ALT), gamma-glutamyl transpeptidase (gamma GTP), glomerular filtration rate, and measurement method of glomerular filtration rate. Table 3 shows the distribution by region of the KOVECO in 2002. In both the veteran group and the second-generation group including each reference, the number of participants is highest in Seoul (26.8% of the veteran group, 25.7% of the second-generation group), followed by Gyeonggi (15.2% and 17.8%) and Busan (9.4% and 7.7%).

## 3. Results

Table 5 shows the distribution of medical facility visit history by year of the KOVECO during the follow-up-period (2002–2018). The prevalence of medical facility visits of the Vietnam War veterans had a gradually increasing tendency, and the average prevalence was 2.10 per year. The prevalence of medical facility visits of the Vietnam War veterans was significantly higher by 11% than those of the reference group in all years in 2002–2018 except 2011 (total prevalence ratio: 1.11; 95% confidence interval (95% CI): 1.1109–1.1142).

## 4. Discussion

The KOVECO represents the whole Korea by using all dispatched troops to the Vietnam War and their second-generation offsprings and the entire Korean general population as reference. It includes various kinds of national representative health information, such as diseases, deaths, and medical examinations, which could be used widely. The data are annually established on December 31 of each year, and it is possible to continuously evaluate the health status of the Vietnam War veterans.

The study is limited by the nature of data from the NHISS. Although most of the veterans are of old age, the data had to be constructed from 2002 to 2018 as a retrospective cohort. Thus, it is difficult for the KOVECO to demonstrate acute or early-onset diseases of the Vietnam War veterans. The diagnosis codes were not sometimes perfectly matched with real diseases because the Korean national health insurance was based on a fee-for-service system [8]. However, the vague disease status is expected to be overcome using this cohort through other information, such as prescriptions, procedures, or inpatient records in a future study.

## 5. Conclusions

The KOVECO is a retrospective cohort to demonstrate the health status of the Korean Vietnam War veterans and their second-generation offsprings. The cohort consists of an annual database and could be updated annually. The cohort includes medical facility visit information, disease or death status, medical examination results, and medical expenses, which could be analyzed in various ways. The benefit of this cohort is the inclusion of the whole Korean population, which can make an extraction of differences of health statuses of the Korean Vietnam War veterans from the general population. The KOVECO could act as a health surveillance system, which would be able to detect the difference between the Vietnam War veteran group and the reference group in unhealthy status described by the prevalence of medical facility visit and provide a direction for policy making through academic research. Furthermore, the KOVECO could describe the health statuses and medical needs of the Vietnam War veterans, and it might be helpful in understanding the health effects of exposure to AO on not only the Korean Vietnam War veterans but the global Vietnam War veterans and citizens who were involved in the Vietnam War.

## Figures and Tables

**Table 1 ijerph-19-04211-t001:** Baseline characteristics of the Vietnam War veteran and second-generation cohort in 2002.

	Vietnam War Veteran		Reference		Second Generation		Reference	
	N	(%)	N	(%)	N	(%)	N	(%)
Total	191,272		1,000,320		377,262		1,963,402	
Male	190,976	(99.8)	998,840	(99.8)	222,389	(58.9)	1,173,061	(59.7)
Female	296	(0.2)	1480	(0.2)	154,873	(41.1)	790,341	(40.3)

**Table 2 ijerph-19-04211-t002:** Distribution by year of the Vietnam War veteran and second-generation cohort during follow-up period.

Year	Vietnam War Veteran	Reference	Total	Second Generation	Reference	Total
2002	191,272	1,000,320	1,191,592	377,262	1,963,402	2,340,664
2003	190,696	988,177	1,178,873	374,724	1,952,094	2,326,818
2004	190,193	976,347	1,166,540	372,332	1,941,740	2,314,072
2005	189,797	964,266	1,154,063	369,898	1,932,997	2,302,895
2006	189,611	952,740	1,142,351	367,753	1,925,204	2,292,957
2007	189,405	941,312	1,130,717	365,398	1,918,802	2,284,200
2008	189,304	929,792	1,119,096	362,857	1,912,283	2,275,140
2009	189,214	918,159	1,107,373	360,244	1,906,333	2,266,577
2010	189,064	906,102	1,095,166	357,348	1,899,631	2,256,979
2011	188,869	893,209	1,082,078	354,125	1,891,888	2,246,013
2012	188,745	880,229	1,068,974	350,858	1,884,912	2,235,770
2013	188,142	865,206	1,053,348	347,296	1,877,446	2,224,742
2014	187,809	852,719	1,040,528	343,902	1,870,625	2,214,527
2015	187,756	838,368	1,026,124	340,420	1,863,553	2,203,973
2016	187,760	823,267	1,011,027	336,696	1,856,044	2,192,740
2017	187,777	807,341	995,118	332,678	1,848,411	2,181,089
2018	187,836	791,014	978,850	328,595	1,840,358	2,168,953
Total (person-year)	3,213,250	15,329,931	18,543,181	6,042,386	32,285,723	38,328,109

**Table 3 ijerph-19-04211-t003:** Distribution by city of the veteran and second-generation cohort in 2002.

City	Vietnam War Veteran	Reference	Total (% of Column)	Second Generation	Reference	Total (% of Column)
Seoul	52,451	266,688	319,139 (26.8)	105,040	496,945	601,985 (25.7)
Busan	19,143	93,234	112,377 (9.4)	34,419	146,156	180,575 (7.7)
Daegu	9042	53,381	62,423 (5.2)	21,792	95,629	117,421 (5.0)
Incheon	9423	49,698	59,121 (5.0)	16,385	93,923	110,308 (4.7)
Gwangju	4786	26,523	31,309 (2.6)	11,349	51,743	63,092 (2.7)
Daejeon	4957	26,386	31,343 (2.6)	10,627	52,676	63,303 (2.7)
Ulsan	3740	19,212	22,952 (1.9)	8742	38,980	47,722 (2.0)
Gyeonggi	32,796	148,894	181,690 (15.2)	60,720	355,175	415,895 (17.8)
Gangwon	6402	31,775	38,177 (3.2)	11,981	63,140	75,121 (3.2)
Chungbuk	4582	27,814	32,396 (2.7)	9184	59,889	69,073 (3.0)
Chungnam	5944	40,558	46,502 (3.9)	10,853	78,727	89,580 (3.8)
Jeonbuk	6999	40,451	47,450 (4.0)	13,903	79,560	93,463 (4.0)
Jeonnam	7785	48,431	56,216 (4.7)	15,227	86,356	101,583 (4.3)
Gyeongbuk	9969	58,011	67,980 (5.7)	21,282	114,964	136,246 (5.8)
Gyeongnam	11,246	58,372	69,618 (5.8)	21,906	127,136	149,042 (6.4)
Jeju	2007	10,892	12,899 (1.1)	3852	22,403	26,255 (1.1)
Total	191,272	1,000,320	1,191,592 (100)	377,262	1,963,402	2,340,664 (100)

**Table 4 ijerph-19-04211-t004:** List of variables in the KOVECO database.

Database	Details
Personal information (10)	Standard year for the eligibility of the insured
	Personal ID
	Sex (male, female)
	Birth date
	District of residence
	Type of eligibility
	Percentile group of income level
	Grade of disability registered
	Type of disability registered
	Type of group (Vietnam War veteran with reference and second generation with reference)
Medical facility visit information (17)	Personal ID
	Key sequence number of claim
	Medical care institution identification number
	The first day of receiving treatment
	Form of claimed bill
	Medical subject code
	Main disease code (Ko)
	Subdisease code
	First date of inpatient
	Route through hospitalization
	Days of receiving medical care
	Days of visit hospitals or inpatient
	Total days of prescription
	Extra rate applied based on claims reviewed
	Amount of medical expenses based on claims reviewed
	Amount of expenses paid by beneficiary based on claims reviewed
	Amount of expenses paid by insurer based on claims reviewed
General health examination information (18)	Date of examination
	Personal ID
	Past history (stroke, cardiovascular diseases, hypertension, diabetes, dyslipidemia, tuberculosis, and others, including cancer)
	Medical treatment history (stroke, cardiovascular diseases, hypertension, diabetes, dyslipidemia, tuberculosis, and others, including cancer)
	Family history (stroke, cardiovascular diseases, hypertension, diabetes, dyslipidemia, tuberculosis, and others, including cancer)
	Health behavior (smoking, alcohol consumption, and exercise level)
	Anthropometry (height, weight, waist circumference, body mass index, visual acuity, hearing acuity)
	Blood pressure (systolic blood pressure, diastolic blood pressure, and pulse rate)
	Urine spot test
	Hemoglobin
	Fasting blood glucose
	Lipid profile (total cholesterol, triglyceride, HDL cholesterol, and calculated LDL cholesterol)
	Serum creatinine
	AST (SGOT)
	ALT (SGPT)
	Gamma GTP
	Glomerular filtration rate
	Measurement method of glomerular filtration rate

**Table 5 ijerph-19-04211-t005:** Distribution of medical facility visit history by year of the Vietnam War veteran and second-generation cohort during follow-up period.

Year	Cases of Vietnam War Veteran	Prevalence of Medical Facility Visit of Vietnam War Veteran	Prevalence Ratio Compared with General Population	95% Confidence Interval
2002	228,857	1.20	1.05	(1.04–1.06)
2003	254,735	1.34	1.06	(1.05–1.07)
2004	281,540	1.48	1.08	(1.07–1.09)
2005	314,949	1.66	1.09	(1.08–1.10)
2006	336,040	1.77	1.09	(1.08–1.10)
2007	359,613	1.90	1.09	(1.08–1.09)
2008	386,402	2.04	1.08	(1.08–1.09)
2009	406,807	2.15	1.08	(1.08–1.09)
2010	424,660	2.25	1.09	(1.08–1.09)
2011	228,857	1.21	0.95	(0.94–0.96)
2012	471,628	2.50	1.12	(1.11–1.12)
2013	481,380	2.56	1.11	(1.10–1.12)
2014	493,512	2.63	1.12	(1.12–1.13)
2015	503,866	2.68	1.12	(1.11–1.12)
2016	519,312	2.77	1.13	(1.12–1.13)
2017	525,993	2.80	1.12	(1.11–1.13)
2018	544,915	2.90	1.13	(1.12–1.14)
Total (person-year)	6,756,385	2.10	1.11	(1.11–1.12)

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
