# Peer review of "Cohort Profile: The Korean Vietnam War Veterans’ Health Study Cohort (KOVECO)"

_ijerph, 2022, doi:10.3390/ijerph19074211_

Round 1

Reviewer 1 Report

Initially, the manuscript is of interest, as it tries to report the effects of war-use toxins up to the third generation in the Vietnamese population. However, in the end, only outcome presented is the frequency of consultations in health care centers.

The cohort is large and the variables recorded are many. The purpose of the report is not clear. If it is to show the organization of the cohort study and tracking form, records and variables, or if they want to show the effects of the exposure. If it were the latter, the manuscript fails to report relevant results. 

The description of the methodology and composition of the cohort is very extensive and detailed, finally showing only one result. I suggest clarifying the objective of the study, which may be: 1) to make known the composition and organization of the cohort or 2) to show relevant results of its follow-up.

I did some comments in the document as well:

Line 42: add the s at the end of time(s)

Table 1: total Numbers are swithched between colums second generation and references

Results: would be useful to show the increased risk of consultation, that is 11%

Author Response

Reviewer #1

We sincerely thank Reviewer #1 for the supportive feedback. All the responses to the comments are answered here. They are labeled in the manuscript.

(R1.1) Page 1, Line 42:  “time” “times”.

Thank you for the comment. We corrected the typo.

(R1.2) Page 2, Table 1: “1,963,402” does no corresponds to the sum. May be 377,262.

We appreciate the comment. We think that there was mistake during calculation. The numbers are revised.

(R1.3) Page 2, Table 1: “377,262” is corresponds to the column second generation. Numbers are switched.

We appreciate the comment. We think that there was mistake during calculation. The numbers are revised.

(R1.4) Page 6, Line 130: It would be useful to show the increased risk of consultation, that is 11%.

Thank you for the comment. We revised the sentence to clarify the meaning.

Reviewer 2 Report

Article Review for “Cohort profile: the Korean Vietnam-War Veterans’ Health 2 Study Cohort (KOVECO)”  

 Overall, this Reviewer found this manuscript to address a very important and relevant subject: the Veteran of the Vietnam War.  The uniqueness about this particular study is that it describes NOT the US Veteran, but the South Korean Veteran.  The Authors are very fortunate to have access to this rich database and have come to some good conclusions. However, this Reviewer feels that so much more can be gathered from this large database with continued analysis.  I recommend that the Authors address each one of the following observations:

Abstract, Line 20: Recommend inserting the word “offspring” following the end of the sentence to better describe the “second-generation.”  The new sentence should read as follows: “. . . the health status of the Korean Vietnam-War veterans and their second-generation offspring.”

Abstract, Line 23: Recommend inserting the word “offspring” following the term “second-generation.”  The new sentence should read as follows: “. . . were all Korean Vietnam-War veterans and the second generation offspring . . .”

Page 1, Line 36:  Recommend that the authors define use a more accurate term than “weeding” when they describe the why Agent Orange was used. A quick reference suggests that AO was used more as a “tactical herbicide” by the U.S. military, rather than the word “weeding.”

Page 1, Line 40: Recommend replacing the word “as” with the word “by” for clarity.  The new sentence should read as follows: “ . . . veterans who have participated in the war by providing medical service . . .”

Page 1, Line 41:  Recommend the rewording the sentence to read as follows for improved clarity. The new sentence should read as follows: “The epidemiologic research has been performed five times in the period . . .”

Page 1, Lines 42 through 44:  It is unclear what the Authors are trying to say in the following sentence:  “Although each study has obtained remarkable results, the outcomes were limited to bring more confidential products due to limited reference groups.”  Recommend the Authors reword the sentence to better enable the Reader to comprehend what the Authors are saying.

Page 2, Line 61:  Recommend the replacement of the word “was” with the word “were” for accuracy.  The new sentence should read as follows: “The NHISS data were merged . . .”

Page 2, Lines 75 through 77:  I find it fascinating that the study group has far fewer females in the Veteran group than in the Offspring group (0.2 percent versus 40.3 percent) but it is understandable.  I do not have any recommendation other than I look forward to some interesting results!

Page 4, Line 94:  Recommend the replacing the number “3” with the word “three” for accuracy.  The new sentence should read as follows: “The KOVECO consists of three databases. . .”

Page 6, Lines 127-129:  The Authors expertly described the 17 variables contained in the database, however their study results only contain findings in the area of prevalence of medical facility visits (“The prevalence of medical facility visits of the  Vietnam-War veterans had a gradually increasing tendency, and the average prevalence was 2.10 per year.”) As a Reader, I was hoping to learn more about any additional findings in the study that addressed some of the other variables.  Were there no other variables that illustrated differences from the non-Veteran, non-Offspring study groups?  If not, then perhaps I would recommend the Authors to state this in the manuscript. However, if there ARE some differences, it would enhance the manuscript tremendously if the Authors addressed it in this section.

Page 6, Lines 144 through 147:  The authors state that one of the limitations of the study was that “The diagnosis codes were not sometimes perfectly matched with real diseases because the Korean national health insurance was based on the fee-for-ser- vice system. However, the vague disease status can be overcome through other infor- mation such as prescriptions, procedures, or inpatient records.”  Although I am glad that the Authors state this as a study limitation, I would recommend a few more sentences that describe their methodology in how they used the prescriptions, procedures or inpatient records to address this limitation.

Page 7, Line 157:  The Authors conclude the manuscript in a manner that summarizes the study well. However, the Reader is left with some gaps in knowledge.  Specifically in line 157 where the Authors state that the relatively poor health outcomes for the Korean Veteran of the US Vietnam War was “. . . caused by the exposure to AO . . .”  The manuscript failed to describe an association between exposure to AO, and a poor health outcome.  Recommend the Authors address the etiology of AO and better describe how the Veteran was exposed to AO during the war. How do these study findings compare to the US Veteran who was exposed to AO during the war?  Were there any other variables that may have been responsible for the poor health outcomes of the Veteran?

Author Response

Reviewer #2

Thank you for the opportunity to revise our manuscript. We appreciate all valuable comments and suggestions from the editor and reviewers. In our belief the manuscript could substantially improve after making revision accordingly. Following this letter are the reviewer’s comments with our responses. The comments are written in black and the paragraphs with blue color are our answers.

Overall, this Reviewer found this manuscript to address a very important and relevant subject: the Veteran of the Vietnam War.  The uniqueness about this particular study is that it describes NOT the US Veteran, but the South Korean Veteran.  The Authors are very fortunate to have access to this rich database and have come to some good conclusions. However, this Reviewer feels that so much more can be gathered from this large database with continued analysis.  I recommend that the Authors address each one of the following observations:

We sincerely thank Reviewer #2 for the supportive feedback. The reviewer observed our manuscript in detailed manner, so the comments were greatly helpful. All the responses to the comments are answered here. They are labeled in the manuscript.

(R2.1) Abstract, Line 20: Recommend inserting the word “offspring” following the end of the sentence to better describe the “second-generation.”  The new sentence should read as follows: “. . . the health status of the Korean Vietnam-War veterans and their second-generation offspring.”

We appreciate the comment. As you noticed, we inserted the word “offsprings” following the term “second-generation”.

(R2.2) Abstract, Line 23: Recommend inserting the word “offspring” following the term “second-generation.”  The new sentence should read as follows: “. . . were all Korean Vietnam-War veterans and the second generation offspring . . .”

Thank you for the comment. As you noticed, we inserted the word “offsprings” following the term “second-generation”.

(R2.3) Page 1, Line 36:  Recommend that the authors define use a more accurate term than “weeding” when they describe the why Agent Orange was used. A quick reference suggests that AO was used more as a “tactical herbicide” by the U.S. military, rather than the word “weeding.”

We appreciate the comment. As you noticed, we revised the word “weeding” to “a tactical herbicide”.

(R2.4) Page 1, Line 40: Recommend replacing the word “as” with the word “by” for clarity.  The new sentence should read as follows: “ . . . veterans who have participated in the war by providing medical service . . .”

Thank you for the comment. We revised the sentences to clarify the meaning.

(R2.5) Page 1, Line 41:  Recommend the rewording the sentence to read as follows for improved clarity. The new sentence should read as follows: “The epidemiologic research has been performed five times in the period . . .”

We appreciate the comment. We revised the sentences to clarify the meaning.

(R2.6) Page 1, Lines 42 through 44:  It is unclear what the Authors are trying to say in the following sentence:  “Although each study has obtained remarkable results, the outcomes were limited to bring more confidential products due to limited reference groups.”  Recommend the Authors reword the sentence to better enable the Reader to comprehend what the Authors are saying.

Thank you for your comment. We revised the sentence as follows to clarify the meaning (Page 1, Lines 42 through 46).

Although each study has obtained remarkable results, they had limitations since the data used by the study were from individual research institutes. Through this study, we tried to establish a cohort with representativeness, reproducibility, and public confidence using national data.

(R2.7) Page 2, Line 61:  Recommend the replacement of the word “was” with the word “were” for accuracy.  The new sentence should read as follows: “The NHISS data were merged . . .”

Thank you for the comment. We changed the sentence to more accurate expression.

(R2.8) Page 2, Lines 75 through 77:  I find it fascinating that the study group has far fewer females in the Veteran group than in the Offspring group (0.2 percent versus 40.3 percent) but it is understandable.  I do not have any recommendation other than I look forward to some interesting results!

We appreciate your compliment. We expect this cohort would be used for valuable researches.

(R1.9) Page 4, Line 94:  Recommend the replacing the number “3” with the word “three” for accuracy.  The new sentence should read as follows: “The KOVECO consists of three databases. . .”

Thank you for the comment. We revised the sentences to clarify the meaning.

(R1.10) Page 6, Lines 127-129:  The Authors expertly described the 17 variables contained in the database, however their study results only contain findings in the area of prevalence of medical facility visits (“The prevalence of medical facility visits of the Vietnam-War veterans had a gradually increasing tendency, and the average prevalence was 2.10 per year.”) As a Reader, I was hoping to learn more about any additional findings in the study that addressed some of the other variables.  Were there no other variables that illustrated differences from the non-Veteran, non-Offspring study groups?  If not, then perhaps I would recommend the Authors to state this in the manuscript. However, if there ARE some differences, it would enhance the manuscript tremendously if the Authors addressed it in this section.

Thank you for the comment. We have described in Table 1 and 3 how the Vietnam-War veteran group and the second-generation group differ by sex and region from each reference group. We have clarified the reason for using a matched reference group in the Materials and Methods section (Page 2, Lines 72-74).

(R1.11) Page 6, Lines 144 through 147:  The authors state that one of the limitations of the study was that “The diagnosis codes were not sometimes perfectly matched with real diseases because the Korean national health insurance was based on the fee-for-service system. However, the vague disease status can be overcome through other information such as prescriptions, procedures, or inpatient records.”  Although I am glad that the Authors state this as a study limitation, I would recommend a few more sentences that describe their methodology in how they used the prescriptions, procedures or inpatient records to address this limitation.

Thank you for your comment. The sentence you have mentioned was written with the intention that using prescriptions, procedures, or patient records could overcome the vague disease status in a future study using this cohort. The sentence was revised as follows to clarify the meaning:

However, the vague disease status is expected to be overcome through other information such as prescriptions, procedures, or inpatient records in the future study using this cohort.

(R1.12) Page 7, Line 157:  The Authors conclude the manuscript in a manner that summarizes the study well. However, the Reader is left with some gaps in knowledge.  Specifically in line 157 where the Authors state that the relatively poor health outcomes for the Korean Veteran of the US Vietnam War was “. . . caused by the exposure to AO . . .”  The manuscript failed to describe an association between exposure to AO, and a poor health outcome.  Recommend the Authors address the etiology of AO and better describe how the Veteran was exposed to AO during the war. How do these study findings compare to the US Veteran who was exposed to AO during the war?  Were there any other variables that may have been responsible for the poor health outcomes of the Veteran?

We agree with your opinion. We have expressed too conclusively. The sentence was revised as follows:

The KOVECO could act as a health surveillance system, which would be able to detect the difference between the Vietnam-War veteran group and the reference group in unhealthy status described by the prevalence of medical facility visit and to provide a direction for policy making through academic research.